# Identification of a New Serovar of *Salmonella enterica* in Mediterranean Buffalo Calves (*Bubalus bubalis*)

**DOI:** 10.3390/ani12020161

**Published:** 2022-01-11

**Authors:** Luisa D’Angelo, Domenico Vecchio, Debora Cozza, Immacolata La Tela, Maria Rosaria Carullo, Ilaria Menozzi, Erika Scaltriti, Stefano Pongolini, Giorgio Galiero, Esterina De Carlo

**Affiliations:** 1National Reference Centre for Hygiene and Technologies of Water Buffalo Farming and Productions, Istituto Zooprofilattico Sperimentale del Mezzogiorno, Via DelleCalabrie 27, 84131 Salerno, Italy; domenico.vecchio@izsmportici.it (D.V.); giorgio.galiero@cert.izsmportici.it (G.G.); esterina.decarlo@cert.izsmportici.it (E.D.C.); 2Regional Center for Salmonella Typing, Via Salute, Istituto Zooprofilattico Sperimentale del Mezzogiorno, 80055 Portici, Italy; debora.cozza@cert.izsmportici.it (D.C.); immacolata.latela@cert.izsmportici.it (I.L.T.); mariarosaria.carullo@cert.izsmportici.it (M.R.C.); 3Risk Analysis and Genomic Epidemiology Unit, Istituto Zooprofilattico Sperimentale della Lombardia e dell’Emilia-Romagna, 43126 Parma, Italy; ilaria.menozzi@izsler.it (I.M.); erika.scaltriti@izsler.it (E.S.); stefano.pongolini@izsler.it (S.P.)

**Keywords:** buffaloes (*Bubalus bubalis*), water buffalo, salmonella, antibiogram, pathogens, diarrhea, gastroenteritis, stool samples, vaccine, welfare

## Abstract

**Simple Summary:**

The presence of *Salmonella enterica* in buffalo (*Bubalus bubalis*) farms is a cause for great concern, as it is responsible for serious economic losses and is a zoonotic agent capable of contaminating the mozzarella production chain. *Salmonella* induced disease mainly affects buffalo calves and is characterized by severe gastrointestinal lesions, profuse diarrhea, severe dehydration, septicemia and death; in adults it can also induce termination of pregnancy. The main source of infection in the herd is asymptomatic elderly animals that release high bacterial charges through the feces. Other sources of infection are contaminated fodder and water; the infection can also be transmitted by rodents, wild birds, insects and humans. There are several serotypes of *Salmonella* reported in the literature capable of infecting the buffalo. This work describes an important outbreak of Salmonellosis that occurred in young subjects of a buffalo farm located in the province of Salerno. It is supported by a new serotype of *Salmonella* so far never described and included in the Kauffmann–White classification.

**Abstract:**

This case report describes for the first-time cases of severe gastroenteritis in water buffalo calves due to a new serovar of *Salmonella enterica*. The study was carried out on fecal matrix collected from live water buffalo calves that showed profuse diarrhea, severe dehydration and fever, exhibiting a systemic course. Culture and molecular investigations identified the pathogens isolated from intestinal contents as two Salmonella serovars, *Salmonella enterica enterica O:35* and a new serovar of *Salmonella enterica*. The isolates showed multi-drug resistance. Timely diagnosis associated with a targeted antimicrobial treatment were found to be sufficient for the survival and recovery of the infected animals. Herd vaccines prepared from isolated pathogens were used to prevent further deaths of the calves.

## 1. Introduction

*Salmonella enterica* found in water buffalo (*Bubalus bubalis*) herds is a matter of concern since it is responsible for serious economic losses in livestock and is a zoonotic agent responsible for foodborne illness [1]. As for bovine calves, Salmonella induced disease in water buffalo calves is characterized by severe gastrointestinal lesions, profuse diarrhea, severe dehydration and death [1]. Acute Salmonellosis generally induces diarrhea and, in the beginning, mucous, later becoming bloody and fibrinous, often containing epithelial casts. Ingestion is the main route of infection, although it can also occur through the mucosa of the upper respiratory tract and conjunctiva. The major source of infection in the herd is represented by asymptomatic older animals shedding heavy loads of bacteria through feces. Other sources of infection are contaminated forages and water, as well as transmission by rodents, wild birds, insects and humans [1,2]. The disease can also cause sudden death without symptoms. Occasionally, the infection is systemic, affecting joints, lungs and/or the central nervous system (CNS) [1]. Moreover, several *Salmonella* serovars seem to be able to infect water buffalo, mainly affecting 1–12-week-old calves, even though reports on Salmonellosis in B. bubalis are scarce [1,3]. Water buffalo calves are more frequently affected by gastroenteritis than bovine calves, with mortality rates as high as 70% in water buffalo species vs. 50% in bovine [1,2,3,4].

### Observed Salmonella Outbreak in a Farm in Southern Italy–Environmental Background, Critical Points and Chronological Table

The animals examined were from a regional herd located on a farm in Southern Italy, in the province of Salerno. The calf farm was made up of single boxes, or 2 rows of 8 single specular boxes, used up to 20 days of age of the animals. Subsequently, the latter were moved to multiple boxes. Calves lived with their mother up to 2 days; they were then placed in a single box and fed with colostrum for up to 10 days, 2 s/day, carrying on until weaning with reconstituted milk powder. The utensils intended for the preparation of feeding calves were handled in order to avoid cross-contamination phenomena, according to detailed procedures set out in the manual of good hygiene practices. For the water supply, well water was used, subjected to biannual microbiological checks in a private laboratory. Continuous sanitation was provided with a UV lamp placed at the point of entry of the water from the well to the main pipe, to which the use of chlorine was cyclically associated. The workers were in possession of a training certificate, with mainly internal courses, biannually supported by the company veterinarian and by the reference zoonome. Two workers were assigned to the veal farmer who took turns to manage it. The isolation of infected animals or suspected of being infected with one of the zoonotic and non-zoonotic diseases was promptly ensured.

On the limits of the farm and in other housing areas, in the areas destined to the breeding of buffaloes, it was possible to have contact with other livestock and with owner and non-owner pets (dogs and cats). Presence of an educational farm with two donkeys, six sheep, birds (geese, hens) was another critical point. Possibility of entry into the farm of wild animals and the presence of a high number of pigeons throughout the company area were very important critical points for health risk and cross-contamination.

In September 2019, two buffalo calf carcasses affected by lethal gastroenteritis and 28 diarrheal buffalo calf feces were analyzed. The microbiological analysis of the intestinal contents and stool matrix exhibited a positivity for *Salmonella I subsp. 35:1* in one calf (ACC. 96582/19) and in fivestool samples (ACC. 101102/19). Other pathogens were isolated in the calf, such as *Escherichia coli* in intestine, mesenteric lymph node, liver and lungs. Enterocolitis, characterized by severe damage of the intestine, ulcers, cirrhotic liver lesions, bilateral interstitial nephritis, severe bilateral acute pneumonia, proliferative endocarditis, emerged from the necropsy of the calf. The treatment carried out on symptomatic calves on evidence of the intestinal pack, on evaluation of the company veterinarian, involved with a broad-spectrum antibiotic therapy. After a few months, the arrival of the vaccine in January for the aforementioned pathogens made it possible to administer a herd vaccine. The administration protocol, to get a better efficiency of the whole herd, provided for treatments to the pregnant buffaloes up to 9 months; calves borne to vaccinated mothers received the vaccine after 8 weeks, while calves borne to unvaccinated mothers received the vaccine after 4 weeks from birth. The re-vaccination for the first calves took place after 3/4 weeks, while for the second ones it took place after 6 weeks. Subsequently, in April 2020 there were many deaths of vaccinated buffalo calves aged around 20 days, showing signs of severe gastroenteritis. Thus, 8 samples of feces from live animals with an age between 15 and 25 days were analyzed. During this process, an additional isolate of *Salmonella* (ACC. 38020/20) was collected.

So, the attention was mainly focused on the calf farm following high calf mortality reports in the year 2020 (mortality 27.7%). In April, there were 11 deaths of buffalo calves aged 20 days, out of the 24 born. The prevalence was 45.8% (Table 1).

## 2. Materials and Methods

The isolation of *Salmonella enterica* in stool samples was performed according to UNI EN ISO 6579-1:2017 [5]. From the initial matrix, 1 g of each sample was weighed and added to 9 mL of BPW pre-enriched solution (buffered peptone water) and incubated between 34 °C and 38 °C for 18 h ± 2 h. After this step, 0.1 mL of solution was inoculated on three equally spaced MSRV agar points (modified semisolid rappaport vassiliadis agar) incubated at 41.5 °C for 24 h ± 3 h. Suspicious positive MSRV agars showed up as areas of turbidity at the injection sites. A bacteriological loop (1 µL) was immersed in the cloudy area and swiped on a selective medium of BSA agar (brilliance salmonella agar) and one of XLD agar (xylose lysine desoxycholate agar) and incubated at 37 °C for 24 h ± 3 h. Salmonella colonies on the XLD agar grew black in the center and surrounded by a transparent reddish halo, while on the BSA agar they grew smooth and with a violet color. The suspect colonies were transferred onto a non-selective medium, TSA agar (tryptic soy agar), and incubated at 37 °C for 24 h ± 3 h. For the serological (Remel™ agglutinating sera, salmonella polyvalent agglutinating aera, polyH–O antisera, for identification of flagellar and somatic antigens) and biochemical tests, (VITEK^®^ 2 system for a rapid and direct identification in micromethod). All incubations were always performed in aerobic conditions.

In addition, the presence of *Escherichia coli*, *Clostridium perfringens* and parasitic (protozoa, trematodes, nematodes and cestodes) research was also determined in order to investigate their association with *Salmonella* spp. *Escherichia coli* and *Clostridium perfringens* were isolated according to the protocol reported by Quinn et al. [2]. Parasitic research was performed according to flotation and sedimentation techniques reported by URQUHART, G.M [6].

Finally, the samples were also investigated for the presence of *Rotavirus* and *Coronavirus* using a specific Elisa kit (test Rota-Corona-k99 Ag-IDEXX, Johnstone, United Kingdom).

### 2.1. Salmonella Regional Typing Centre

In September 2019, two strains from a Salernitan buffalo company were typed by the Salmonella Typing Centre.

The Serotyping was carried out searching for the somatic antigens by rapid sero-agglutination test on a slide (sieriStatens Serum Institut) and the flagellar by slow agglutination in a test tube (DIFCO antisieri). After an analysis of serological grouping, they were found to belong to group O:35, with one of the second flagellar phases unexpressed. (*S.enterica sub. enterica 35:lw:-*).

In April 2020, another strain that showed the same antigenic profile was isolated in the same company. In an attempt to research the unexpressed flagellar phase, it was noted that it existed, but that it was none of the ones already described in the Kauffmann–White handbook (*S.entericasub.entericaO35:lw:z41*).

Repeating the research of the flagellar antigens on the previously isolated, stored in strain collection strains, it was noted that the same serotype had already been isolated in September 2019.

### 2.2. Susceptibility Test

The antibiotic resistance study was carried out through microdilution in broth following the Sensititre System method and using the EU Directive 2013/652/eu of the 12 November 2013 for the interpretation of the results.

The strains underwent tests for antibiotic resistance showing sensibility towards the molecules tested: cefotaxime-ceftazidime-ciprofloxacina–cloramfenicolo-gentamicina-tetraciclina-sulfametossazolo-tigeciclina-trimethoprim-colistina-ac.nalidixico-ampicillina-azitromicina-meropenem. The bacterial resistance found instead: erythromycin- lincomycin- spectinomycin- neomycin-pennicillin-rifampicin-spiramycin-tilmycosin.

### 2.3. Whole Genome Sequencing

All three isolates of *S. entericasub. enterica serovar 35:lw:z41* (96582/19, 101102/19 and 38020/20) underwent whole genome sequencing (WGS) to identify genomic characteristics. Data are available at EBI under study accession n. PRJEB46074.

Whole genome libraries were prepared using Illumina DNA Prep (M) TagmentationKit (Illumina, San Diego, CA, USA) starting from genomic DNA extracted by Qiagen DNA purification kit according to the manufacturer’s instructions (Qiagen, Milan, Italy). An Illumina MiSeq platform (Illumina, San Diego, CA, USA) was used to obtain raw sequencing reads (250 × 2 bp paired-end) that were checked for quality using FastQC (https://www.bioinformatics.babraham.ac.uk/projects/fastqc/) (accessed on 15 September 2021). Bacterial species confirmation was carried out using Kraken2 software. Reads were trimmed with Trimmomatic ver. 0.38 according to [7] and assembled using Unicyclerver0.4.8 [8].

All three isolates belong to Sequence Type (ST)8452 (aroC 991, dnaN 14, hemD 108, hisD 1596, purE 6, sucA 1177, thrA 104) determined by the insilico multi locus sequence typing tool powered byInstitute Pasteur (BIGSdb for *Salmonella enterica*: https://pubmlst.org/bigsdb?db=pubmlst_salmonella_seqdef) (accessed on 22 September 2021).

Antimicrobial resistance genes were detected using the ResFinder-3.2 (https://cge.cbs.dtu.dk/services/ResFinder/) (accessed on 22 September 2021). database detecting an aminoglycoside resistance gene, namely aac(6′)-Iaa, while virulence genes, identified using the VF analyzer Tool of Virulence Factor Database (VFDB, http://www.mgc.ac.cn/) (accessed on 28 September 2021), detected genes listed below: fimbrial(*agf*, *csg*, *bcf*, *fim*, *lpf*, *peg*, *sta*, *stb*, *std*, *ste*, *stf*, *sth*, *sti*) and nonfimbrial (*misL*, *ratB*, *shdA*, *sinH*) adherence determinants, secretion system (TTSSencoded by SPI-1 and SPI-2, TTSS translocated effectors), afimbrial adhesin AFA-1(*afaB* and *afaC*), autotransporter (*ehaB*), PhoPQ regulation system (*phoP* and phoQ), Macrophage-inducible (*mig-14*)genes. No plasmid was detected using PlasmidFinder 2.1 software (https://cge.cbs.dtu.dk/services/PlasmidFinder/) (accessed on 24 September 2021).

Insilico serotyping was performed using SeqSero2 software [9], identifying a rare antigenic formula (35:l,w:z41) that corresponds to the antigenic formula detected by seroagglutination. A serovar reference strain, namely SAL_GB9529AAwith antigenic formula 35:l,w:z41, was deposited in Enterobase since march 2021. In addition, this reference strain originates from the farm of the study and was deposited by Institute Pasteur, where it was sent for confirmation.

All three isolate were deposited in Enterobase Salmonella database [10,11] under the IDs SAL_JB2510AA, SAL_JB2511AA, SAL_JB2512AA, respectively, for 96582/19, 101102/19 and 38020/20isolates. Core genome sequence type (cgST) was determined using the EnterobasecgMLSTAnalysistool through thehierCC (hierarchical clustering of CgMLST) method [10,11] and visualized using minimum spanning tree (MST).

As shown in Figure 1, the EnterobasecgMLST analysis revealed that 96582/19 and 101102/19 isolates differ from each other byone allele and from the SAL_GB9529AA isolate by one and two alleles, respectively, sharing the same HC2cgST (258938). Moreover, our last isolated strain (38020/20) differs from the others by two and three alleles sharing the same HC5 cgST (258938).

The limited observed variability among the isolates is compatible with genomic diversity that can originate from the conditions of farm contamination where multiple successive infections occur over time.

## 3. Results

*S. entericasub. enterica serovar35:lw:z41* was isolated in two of eight stool samples tested. The limited observed variability among the isolates is compatible with genomic diversity that can originate from the conditions of farm contamination where multiple successive infections occur over time. In addition, *haemolytic Escherichia coli* have also been found, while *Clostridium perfringens*, parasitic agents, *Rotavirus* and *Coronavirus* were negative.

## 4. Discussion

The administration of antibiotic therapy allowed symptomatic resolution in debilitated subject. With the use of the housing vaccine, it also contained the number of deaths of symptomatic animals. In contrast to an antibiotic therapy, where the benefits are limited to the individual taking the drug, vaccines have the potential for far-reaching effects that include health, animal welfare, and, ultimately, livestock productivity. Furthermore, external and internal biosecurity measures, hygiene management of litter material, daily use tools, adsorbents, correct milking hygiene, evaluation of the teat score and, above all, good management of the dry, were the main points for the pathogen control [12]. The discovery of *S. enterica sub. enterica serovar 35:lw:z41* lead to an assessment of its presence, if any, in the surrounding farms in order to understand the actual source of contamination, for example, incubator animals or cross-contamination in livestock practices.

## 5. Conclusions

Finally, the spread of this serotype in other reservoir animals remains to be investigated. This also applies to the clarification of its zoonotic potential, since it has not yet been reported in humans; therefore it is necessary to wait for further future epidemiological data to evaluate the presence of any reservoirs in addition to the *Salmonella enterica buffalo O35:lw:z41* and its ability to infect and give disease in humans.

## Figures and Tables

**Figure 1 animals-12-00161-f001:**
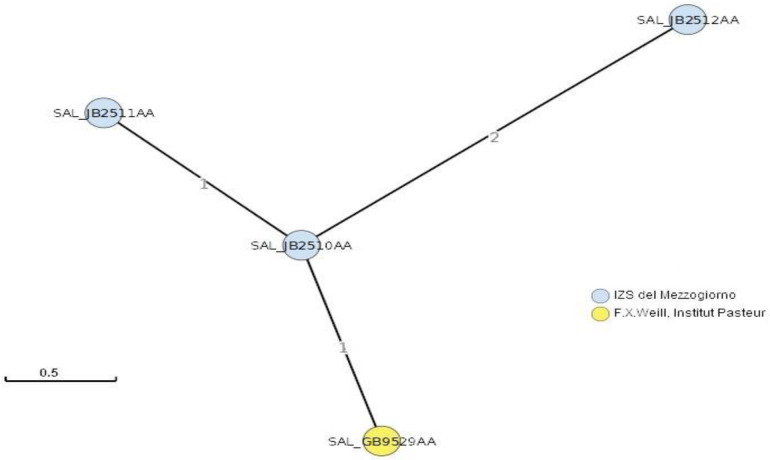
Minimum spanning tree (MST) based on cgMLST. In blue are highlighted SAL_JB2510AA, SAL_JB2511AA, SAL_JB2512AA isolates (respectively corresponding to 96582/19, 101102/19 and 38020/20), while in yellowSAL_GB9529AA isolate (deposited by Institute Pasteur). Numbers on branches indicate the allelic distance (AD) among connected isolates.

**Table 1 animals-12-00161-t001:** Mortality year 2020.

2020	Births	Deaths	Age at Death (Days)	Prevalence (%)
January	14	4	33	28.5
February	14	9	31.5	64.2
March	10	3	33.66	30
**April**	**24**	**11**	**20.09**	**45.8**
May	8	1	1	12.5
June	15	2	1	13.3
July	17	0	0	0
TOT	102	30	17.17	27.75

## Data Availability

Not applicable.

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
