# Peer review of "Identification of a New Serovar of Salmonella enterica in Mediterranean Buffalo Calves (Bubalus bubalis)"

_animals, 2022, doi:10.3390/ani12020161_

Round 1

Reviewer 1 Report

The research topic is relevant and important since authors claim they have isolated a novel Salmonella serovar. The authors have conducted the whole genome sequencing and antibiotics resistance profiling. However, all the results are not shown. More data regarding the resistance profile is needed. The manuscript sound to be written by non-native English speakers, so extensive English language editing and styling is required.

More results on virulence typing, genotypic AMR profiling and serotyping is required. In addition, below are some the common mistakes found in the manuscript.

The scientific names should be italicized.

Proper English is not used in many places and thorough revision is needed. Typos such as "acute Salmonellas" instead of "acute salmonellosis" on line 43 should be fixed. 

Decimal points should be separated by periods, not commas. Appropriate units should be used. For eg, line 107 1 g instead of 1gr.34°C instead of 34 ° C ., 1 ml instead of 1ml etc.

Line 123-124. The presence of Escherichia coli, Clostridium perfringens and parasitic research...... Not clear what is meant by parasitic research. The sentence doesn't flow.

Some paragraphs are extremely short as one sentence.

Author Response

Thank you very much for the recommended considerations.

Please, consider the version I attach, as the latest revised and corrected version.

My greetings

Luisa

Reviewer 2 Report

The authors report the identification of a new Salmonella enterica serovar in buffalo calves. The manuscript requires some revision before its acceptance for publication.

Simple summary:

Line 14, … in buffalo (Bubalus bubalis) farms …

Lien 20, … and water, and the infection can also be transmitted by rodents, …

Introduction

Line 41, A Salmonella-induced disease

Line 43, Acute Salmonellosis generally induces

Line 44, mucous in the beginning, …

Line 48 … water and transmission by

Line 56 and 79, I would recommend to put this in one chapter entitled e.g. “Observed Salmonella outbreak in a farm in Southern Italy – environmental background, critical points and chronological table”

Line 60, the first part of the sentence is fragmentary.

Line 64, can a reference be given for the manual of good hygiene practices?

Line 70, took turns

Line 102, was 45. 8%

Materials and Methods

Line 124, parasitic research?

Lines 126-127, not a full sentence

Line 128, samples were also investigated for the presence of

Line 138, 2020, another strain was isolated in the same company

Discussion

Line 207, In contrast to an antibiotic therapy where the its benefits are limited to …, vaccines …

Conclusions

Line 217, Finally, the spread of this serotype in … remains to be investigated. This also applies to the clarification of its …

Author Response

Please,

observe the correct article attached.

Thank you

Luisa

Round 2

Reviewer 1 Report

You don't capitalize S in "salmonellosis" when it is not starting a sentence.